# Early and Long-Term Results of Simultaneous and Staged Revascularization of Coronary and Carotid Arteries

Elena Golukhova, Igor Sigaev, Milena Keren *, Inessa Slivneva *, Bektur Berdibekov, Nina Sheikina, Olga Kozlova, Valery Arakelyan, Irina Volkovskaya, Tatiana Zavalikhina and Susanna Avakova

A.N. Bakulev National Medical Scientific Center for Cardiovascular Surgery, 121552 Moscow, Russia; ezgolukhova@bakulev.ru (E.G.); iysigaev@bakulev.ru (I.S.); bsberdibekov@bakulev.ru (B.B.); n9623951466@gmail.com (N.S.); oskozlova@bakulev.ru (O.K.); vsarakelyan@bakulev.ru (V.A.); ivvolkovskaya@bakulev.ru (I.V.); tzavalikhina@bakulev.ru (T.Z.); saavakova@bakulev.ru (S.A.)
* Correspondence: makeren@bakulev.ru (M.K.); ivslivneva@bakulev.ru (I.S.)

**Abstract:** Background: Carotid artery disease is prevalent among patients with coronary heart disease. The concomitant severe lesions in the carotid and coronary arteries may necessitate either simultaneous or staged revascularization involving coronary bypass and carotid endarterectomy. However, there is presently a lack of consensus on the optimal choice of surgical treatment tactics for patients with significant stenoses in both carotid and coronary arteries. The aim of the current study was to compare the 30-day and long-term outcomes of coronary and carotid artery revascularization surgery based on the simultaneous or staged surgical tactics. Material and Methods: This single-center retrospective study involved 192 patients with concurrent coronary artery disease and carotid artery stenosis $\geq$ 70%, of whom 106 patients underwent simultaneous intervention (CABG + CEA) and 86 patients underwent staged CABG/CEA. The mean time between stages ranged from 1 to 4 months (mean $1.88 \pm 0.9$ months). The endpoints included death from any cause, non-fatal stroke, non-fatal myocardial infarction (MI), and major adverse cardiovascular events (MACEs) (death + non-fatal MI + non-fatal stroke) within 30 days after the last intervention and in the long-term follow-up period (median follow-up—6 years). Results: The 30-day all-cause mortality, incidence of postoperative non-fatal MI, non-fatal stroke, and MACEs did not exhibit differences between the groups after single-stage and staged interventions. However, the overall risk of postoperative complications (adjusted for the risk of any complication per patient) (OR 2.214, 95% CI 1.048–4.674, $p = 0.035$), as well as the duration of ventilatory support ($p = 0.004$), was elevated in the group after simultaneous interventions compared with the staged intervention group. This difference did not result in an increased incidence of death and MACEs in the group after simultaneous interventions. In the long-term follow-up period, there were no significant differences observed when comparing simultaneous or staged surgical tactics in terms of overall survival (54.9% and 62.6% in Groups 1 and 2, respectively, P log-rank = 0.068), non-fatal stroke-free survival (45.6% and 33.6% in Groups 1 and 2, respectively, P log-rank = 0.364), non-fatal MI-survival (57.6% and 73.5% in Groups 1 and 2, respectively, P log-rank = 0.169), and MACE-free survival (7.1% and 30.2% in Groups 1 and 2, respectively, P log-rank = 0.060). The risk factors associated with an unfavorable outcome included age, smoking, BMI, LV EF, and atherosclerosis of the lower extremity arteries. Conclusions: This study revealed no significant difference in the impact of simultaneous CABG + CEA or staged CABG/CEA on the incidence of death, stroke, MI, and MACEs over a 30-day and long-term follow-up period. Although the immediate results indicated an increased risk of a complicated course (attributable to overall complications) and more prolonged ventilation after simultaneous CABG + CEA compared with staged CABG/CEA, this did not lead to an increase in fatal complications. Therefore, the implementation of either tactic is considered eligible and appropriate following a thorough operative risk assessment.

**Keywords:** coronary artery disease; carotid stenosis; coronary artery bypass grafting; carotid endarterectomy

## 1. Introduction

The prevalence of carotid artery disease among patients with coronary heart disease (CHD) ranges from 5 to 9% [1]. In a study conducted by Naylor et al., the prevalence of carotid artery stenosis exceeding 50% among patients who underwent coronary artery bypass grafting (CABG) was 9%, and for stenosis exceeding 80% it was 7% [2]. The presence of critical lesions in both vascular areas necessitates the selection of optimal treatment strategies, which may involve invasive interventions in addition to medical treatment. These interventions can be performed through surgical, endovascular, or hybrid approaches.

When indications for open surgical intervention are present, potential treatment tactics include simultaneous or staged coronary artery bypass grafting (CABG) and carotid endarterectomy (CEA). However, the choice of optimal surgical tactics remains debatable and is primarily influenced by local management protocols for such patients. The current situation arises from contradictory data on the immediate and long-term outcomes of surgery obtained from observational and randomized studies and meta-analyses. These data support both simultaneous [3,4] and staged intervention tactics [5,6], preventing the formulation of unequivocal recommendations. Clinical guidelines, on the other hand, offer mainly general recommendations for managing patients with coronary and carotid stenosis, emphasizing the need for individualizing each patient's approach, considering the potential risks and benefits of the intervention.

The objective of our study was to compare the 30-day and long-term outcomes of coronary and carotid artery revascularization surgery based on the selected simultaneous or staged surgical strategies.

## 2. Materials and Methods

A single-center retrospective study was conducted to evaluate the 30-day and long-term outcomes (with a follow-up period extending up to 9 years and a median follow-up of 6 years) in patients with severe coronary and carotid artery disease who underwent both CABG and CEA via either a simultaneous or two-stage intervention. Patients were stratified based on the surgical approach employed. The decision for intervention was reached through a consensus among a multidisciplinary team. In the simultaneous intervention, both CABG and CEA were performed under a single endotracheal anaesthesia. For staged interventions, the first stage was always CABG, and the second stage was CEA; the average time between stages ranged from 1 to a maximum of 4 months (mean $1.88 \pm 0.9$ months).

Inclusion criteria encompassed patients with severe CHD concurrent with severe stenosis of the carotid arteries ($\geq$70% as per European Carotid Surgery Trial (ECST) ultrasound criteria [7], including stenosis assessment by lesion area). Eligible patients were those recommended for either simultaneous intervention (CABG + unilateral CEA) or a staged approach (stage 1—CABG, stage 2—unilateral CEA).

Patients with left ventricular ejection fraction (LV EF) less than 40%, a history of previous surgical and/or endovascular interventions on the heart and carotid arteries (such as re-CABG, coronary or carotid artery stenting), valve pathology requiring surgical intervention, or the presence of LV aneurysm requiring surgical correction were excluded from the study. Additionally, patients recommended to undergo CEA first, followed by CABG, were also excluded.

Study endpoints included all-cause death, non-fatal stroke, non-fatal myocardial infarction (MI), and major adverse cardiovascular events (MACEs) (death + non-fatal MI + non-fatal stroke). Additionally, we evaluated other postoperative complications occurring within 30 days and assessed the associated risks. Predictors for both 30-day MACEs and long-term endpoints were also calculated.

Data analysis involved electronic medical records documented between 2013 and 2018. The primary dataset comprised 3844 patients with severe coronary and carotid pathology. Following the application of predefined inclusion and exclusion criteria, 392 patients were identified from the initial cohort. Subsequently, 66 individuals were excluded due to incomplete primary data, resulting in a final sample of 326 patients. Among these, long-

term outcomes could be tracked for 192 patients, forming two distinct follow-up groups. Thus, Group 1 included 106 patients who underwent a simultaneous intervention with CABG + CEA, and Group 2 included 86 patients who underwent a two-stage intervention (stage 1—CABG, stage 2—CEA).

All patients underwent ultrasound examination of the carotid arteries, conducted using the ECST evaluation criteria which included an assessment of the stenosis area and an estimation of linear blood flow velocity in the stenosis area [7] and transthoracic echocardiography for the assessment of LV EF using the Simpson's biplane method [8]. Furthermore, ultrasound and angiographic assessments were performed to evaluate the patency of renal arteries and lower extremities arteries. Coronarography was conducted to assess the severity of coronary lesions.

Clinical characteristics, preoperative examination results, and the structure of surgical interventions among the studied groups are presented in Table 1. The patients in both groups exhibited no significant differences in terms of age, sex, and most clinical characteristics. However, patients who underwent the simultaneous CABG + CEA procedure had a lower BMI and a higher prevalence of smoking. The analysis of preoperative data showed the initial severity of patients' condition in both groups, attributed to significant comorbidity (history of MI, hypertension, heart failure, diabetes mellitus, chronic obstructive pulmonary disease (COPD), CHD, etc.), severe coronary artery disease, including stenosis of the left coronary artery trunk, and concomitant lesions of other vascular system (renal arteries, lower extremity lesion). The proportion of symptomatic patients who had previously experienced a stroke was identical in both groups, amounting to 19.8%.

**Table 1.** Clinical characteristics of the study groups.

| Variables | Group 1 | Group 2 | *p*-Value |
|---|---|---|---|
| | CABG + CEA (n = 106) | CABG/CEA (n = 86) | |
| *Anthropometric data* | | | |
| Gender, n (%) | M—90 (84.9) F—16 (15.1) | M—66 (76.7) F—20 (23.3) | 0.150 |
| Age, years, Me [IQR] | 64 [60–69] | 66 [61–69] | 0.115 |
| BMI, kg/m$^2$, Me [M ± SD] | 28.20 ± 3.68 | 29.35 ± 4.09 | 0.041 * |
| *Clinical and anamnestic data, risk factors, comorbidity* | | | |
| III-IV angina (CCS), n (%) | 91 (85.8) 15 (14.2) | 78 (90.7) 8 (9.3) | 0.304 |
| Previous MI, n (%) | 44 (41.5) | 36 (41.9) | 0.961 |
| Previous Stroke, n (%) | 21 (19.8) | 17 (19.8) | 0.994 |
| Hypertension, n (%) | 106 (100) | 84 (97.7) | 0.199 |
| Smoking, n (%) | 46 (43.4) | 20 (23.3) | 0.003 * |
| CKD, n (%) | 24 (24.4) | 16 (18.6) | 0.493 |
| Diabetes, n (%) | 22 (20.8) | 28 (32.6) | 0.064 |
| COPD, n (%) | 10 (9.5) | 2 (2.3) | 0.069 |
| *Diagnostic data* | | | |
| LV EF, %, Me [IQR] | 55 [52–55] | 55 [51–56] | 0.090 |
| One-vessel disease, n (%) | 5 (4.7) | 2 (2.3) | 0.463 |
| Two-vessel disease, n (%) | 24 (22.6) | 15 (17.4) | 0.373 |
| Three-vessel disease, n (%) | 77 (72.6) | 68 (79.1) | 0.303 |
| LCA trunk disease > 50%, n (%) | 40 (38.1) | 25 (29.4) | 0.207 |
| Target CA stenosis, %, Me [IQR] | 77 [70–90] | 77 [70–87] | 0.387 |

**Table 1.** *Cont.*

| Variables | Group 1 CABG + CEA (n = 106) | Group 2 CABG/CEA (n = 86) | *p*-Value |
|---|---|---|---|
| Linear velocity of the blood flow in target CA, cm/s, Me [IQR] | 247 [187–279] | 246 [208–291] | 0.848 |
| Bilateral CA stenosis > 70%, n (%) | 10 (9.4) | 8 (9.3) | 1.0 |
| Stenosis of the renal arteries > 70%, n (%) | 2 (1.9) | 0 | 0.503 |
| Lower extremity artery stenosis > 70%, n (%) | 9 (8.5) | 6 (6.9) | 0.791 |
| *Interventions data* | | | |
| CABG off-pump, n (%) | 32 (30.2) | 26 (30.2) | 0.995 |
| CABG CPB, n (%) | 74 (69.8) | 60 (69.8) | |
| LITA to the LAD, n (%) | 83 (78.3) | 76 (88.4) | 0.066 |
| CPB time, min, [M ± SD] | 70.3 ± 12.4 | 72.4 ± 11.9 | 0.788 |
| Eversion CEA, n (%) | 65 (61.3) | 49 (57.0) | 0.542 |
| CEA with a synthetic patch angioplasty, n (%) | 41 (38.7) | 37 (43.0) | |

\*—differences in indicators are statistically significant ($p < 0.05$). CABG—coronary artery bypass grafting, CEA—carotid endarterectomy, BMI—body mass index, CCS—Canadian Cardiovascular Society, MI—myocardial infarction, AF—atrial fibrillation, CKD—chronic kidney disease, GFR—glomerular filtration rate, COPD—chronic obstructive pulmonary disease, LV EF—left ventricular ejection fraction, LCA—left coronary artery, CA—carotid artery, CPB—cardiopulmonary bypass, LITA—left internal thoracic artery, LAD—left anterior descending artery.

CABG was performed in both groups with either cardiopulmonary bypass or off-pump CABG. CEA was carried out using either the eversion method or a synthetic patch (Table 1).

### 3. Statistical Analysis

Statistical analysis was performed using SPSS 29.0.0 (IBM, Armonk, NY, USA) and StatTech v. 3.1.8 (StatTech LLC, Kazan, Russia). The normal distribution of quantitative indicators was assessed with the Kolmogorov–Smirnov criterion (applied when the number of subjects exceeded 50. In instances of non-normal distribution, quantitative data were described using the median (Me) along with the lower and upper quartiles [Q1–Q3]. Categorical data were described with absolute values and percentages. Comparison of two groups for a normally distributed quantitative indicator with equal variance was carried out using Student's t-criterion. For a non-normally distributed quantitative indicator, the Mann–Whitney U-criterion was employed. Patient survival function was evaluated using the Kaplan–Meier method, and patient survival was analyzed with the Cox regression method. Binary logistic regression analysis was conducted to predict 30-day outcomes. Statistical significance was considered at a level of $p < 0.05$.

### 4. Results

*4.1. 30-Day Outcomes of Simultaneous and Staged CABG and CEA*

The 30-day all-cause mortality, incidence of postoperative non-fatal MI, non-fatal stroke, and MACEs did not differ between the groups. Additionally, there were no differences between the groups regarding other postoperative complications (Table 2), except for a longer duration of mechanical ventilation in the group after simultaneous interventions compared to staged interventions ($p = 0.004$). The risk of postoperative complications (considering the risk of any complication per patient) was higher after simultaneous CABG + CEA (OR 2.214, 95% CI 1.048–4.674, $p = 0.035$) than in the group after staged CABG/CEA, although this did not result in an increased incidence of death and MACEs in the group after simultaneous interventions (Table 2). The risk factors for 30-day MACEs are outlined in Table 3.

**Table 2.** Thirty day complications.

| Complications | Group 1 | Group 2 | OR | 95% CI | *p*-Value |
| --- | --- | --- | --- | --- | --- |
| | CABG + CEA (n = 106) | CABG/CEA (n = 86) | | | |
| *Complications* | | | | | |
| Death, n (%) | 3 (2.8) | 2 (2.3) | 1.233 | 0.133–5.006 | 1.0 |
| Non-fatal MI, n (%) | 0 | 2 (2.3) | 6.302 | 0.299–133.032 | 0.199 |
| Non-fatal stroke, n (%) | 3 (2.8) | 1 (1.2) | 2.476 | 0.041–3.954 | 0.629 |
| MACEs, n (%) | 6 (5.7) | 5 (5.8) | 1.029 | 0.303–3.493 | 1.0 |
| Brain swelling, n (%) | 2 (1.9) | 0 | 4.139 | 0.011–5.100 | 0.503 |
| GI bleed, n (%) | 1 (0.9) | 0 | 2.406 | 0.016–10.106 | 1.0 |
| Respiratory failure, n (%) | 2 (1.9) | 1 (1.2) | 1.635 | 0.055–6.863 | 1.0 |
| Heart rhythm disorders, n (%) | 8 (7.5) | 4 (4.7) | 1.673 | 0.174–2.056 | 0.553 |
| Acute heart failure, n (%) | 2 (1.9) | 2 (2.3) | 1.238 | 0.171–8.976 | 1.0 |
| Multiple organ failure, n (%) | 2 (1.9) | 2 (2.3) | 1.238 | 0.171–8.976 | 1.0 |
| Duration of mechanical ventilation, h, Me [IQR] | 15.5 [10.0–18.1] | 11.5 [8.0–16.2] | | | 0.004 * |
| Rethoracotomy for bleeding, n (%) | 7 (6.6) | 6 (7.0) | 1.061 | 0.343–3.282 | 1.0 |
| Infectious complications, n (%) | 7 (6.6) | 2 (2.3) | 2.970 | 0.068–1.665 | 0.191 |
| *Risk of complications* | | | | | |
| Number of patients with 1 or more any postoperative complication, n (%) | 28 (26.4) | 12 (14.0) | 2.214 | 1.048–4.674 | 0.035 * |

*—differences in indicators are statistically significant ($p < 0.05$). CABG—coronary artery bypass grafting, CEA—carotid endarterectomy, MI—myocardial. infarction, MACEs—major adverse cardiac events, GI—gastrointestinal tract. Risk factors for the occurrence of 30-day MACEs were identified using binary logistic regression (Table 3). According to the univariate analysis, variables such as age and BMI were associated with 30-day postoperative MACEs. However, in multivariate regression analysis, only age emerged as an independent predictor linked to the onset of MACEs (OR: 1.15; 95% CI: 1.03–1.27; $p = 0.01$).

**Table 3.** Risk factors for 30-day MACEs.

| Risk Factors | Univariate Analysis | | Multivariate Analysis | |
| --- | --- | --- | --- | --- |
| | OR (95% CI) | *p*-Value | OR (95% CI) | *p*-Value |
| Age per 1 y | 1.16 (0.05–1.28) | 0.004 * | 1.15 (1.03–1.27) | 0.01 * |
| Female | 1.68 (0.42–6.68) | 0.460 | - | - |
| BMI per 1 kg/m$^2$ | 0.81 (0.66–0.97) | 0.023 * | 0.81 (0.66–1.01) | 0.057 |
| Smoking | 1.64 (0.48–5.58) | 0.429 | - | - |
| Previous stroke | 2.47 (0.68–8.92) | 0.167 | - | - |
| Previous MI | 2.59 (0.73–9.16) | 0.140 | | |
| Bilateral CA stenosis | 0.96 (0.12–8.00) | 0.973 | - | - |
| LV EF | 0.94 (0.71–1.64) | 0.647 | - | - |
| Three-vessel disease | 3.41 (0.42–27.35) | 0.249 | - | - |
| LCA trunk disease > 50% | 0.42 (0.09–1.99) | 0.272 | - | - |
| Lower extremity artery stenosis | 1.19 (0.14–10.00) | 0.871 | - | - |
| Diabetes | 1.07 (0.27–4.20) | 0.924 | - | - |
| CKD | 0.36 (0.04–2.93) | 0.342 | - | - |
| COPD | 3.80 (0.72–19.98) | 0.115 | - | - |

**Table 3.** *Cont.*

| Risk Factors | Univariate Analysis | | Multivariate Analysis | |
|---|---|---|---|---|
| | OR (95% CI) | *p*-Value | OR (95% CI) | *p*-Value |
| CABG CPB | 1.16 (0.30–4.55) | 0.827 | - | - |
| Stage | 1.03 (0.31–3.49) | 0.964 | - | - |

*—differences in indicators are statistically significant ($p < 0.05$). MACEs—major adverse cardiac events, BMI—body mass index, MI—myocardial infarction, CA—carotid artery, LV EF—left ventricular ejection fraction, LCA—left coronary artery, CKD—chronic kidney disease, COPD—chronic obstructive pulmonary disease, CABG—coronary artery bypass grafting, CPB—cardiopulmonary bypass.

### 4.2. Long-Term Results

The median follow-up in the overall patient cohort was 6 [5–8] years, with 6 [5–8] years for simultaneous interventions and 7 [5–9] years for staged interventions. Throughout the follow-up period, 33 deaths occurred after simultaneous interventions and 26 deaths after staged interventions; the overall survival rates were 54.9% and 62.6% in Groups 1 and 2, respectively (P log-rank = 0.068) (Figure 1). Non-fatal strokes occurred in 25 patients after simultaneous interventions and in 23 patients after staged interventions; stroke-free rates were 45.6% and 33.6% in Groups 1 and 2, respectively (P log-rank = 0.364) (Figure 2). Non-fatal myocardial infarctions occurred in 21 patients in the simultaneous intervention group and 17 patients in the staged intervention group; for the MI, the event-free rates were 57.6% and 73.5% in Groups 1 and 2, respectively (P log-rank = 0.169) (Figure 3).

MACEs were observed in 79 patients in the group following simultaneous interventions and in 66 patients after staged interventions, with MACE-free rates of 7.1% and 30.2%, respectively (P log-rank = 0.060) (Figure 4). The impact of risk factors on outcomes was evaluated for each of the distant endpoints. The univariate Cox analysis revealed that parameters such as BMI and smoking were associated with the risk of all-cause mortality. In multivariate Cox analysis, BMI emerged as an independent risk factor for all-cause mortality. It is noteworthy that there was an inverse dependence between the BMI and the outcome: a 1 kg/m² increase in BMI was associated with a decreased risk of all-cause mortality (HR 0.93, 95% CI 0.85–0.99, $p = 0.036$) (Table 4).

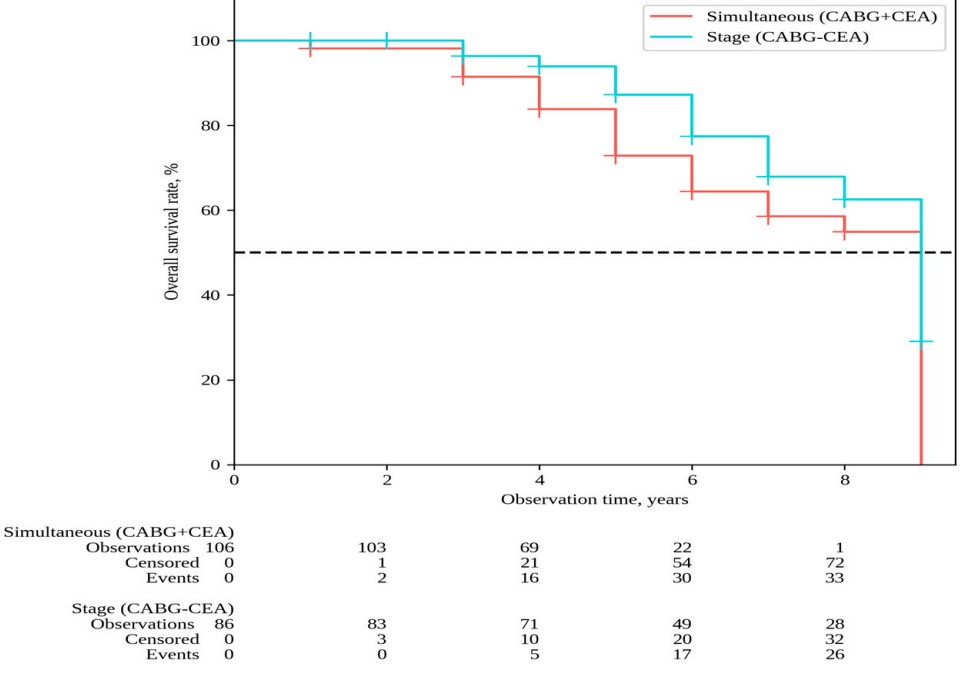

**Figure 1.** Overall survival.

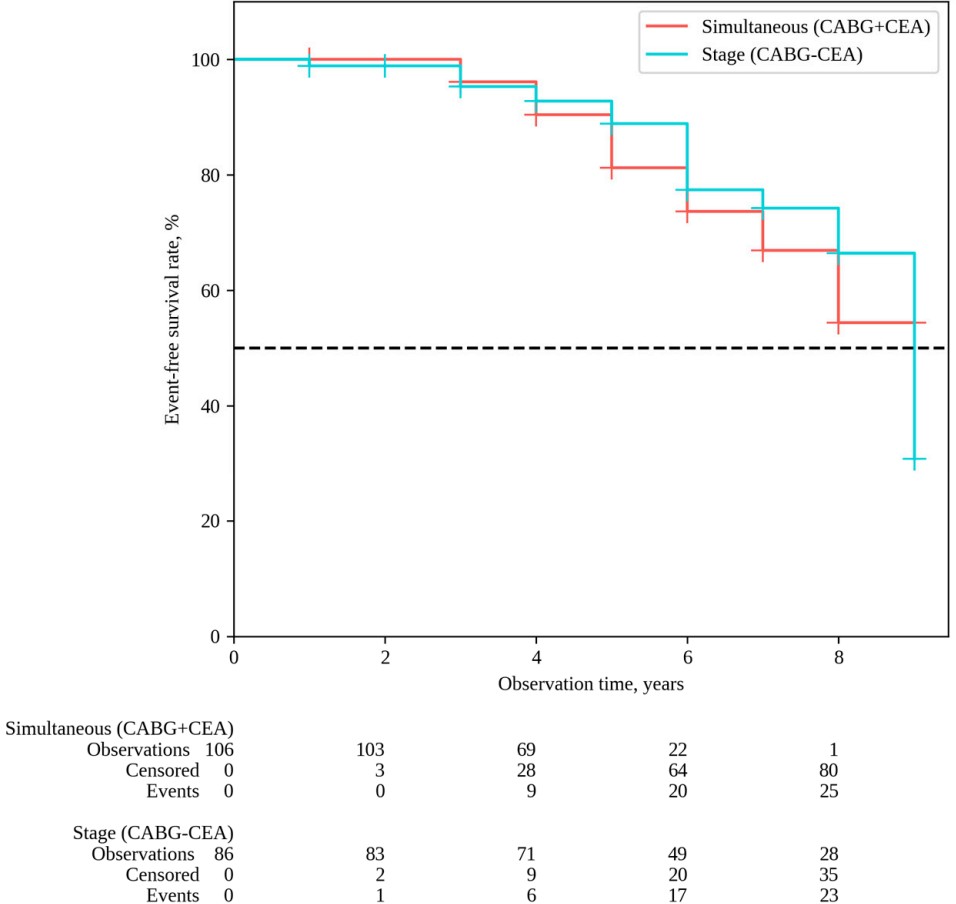

Figure 2. Stroke-free survival.

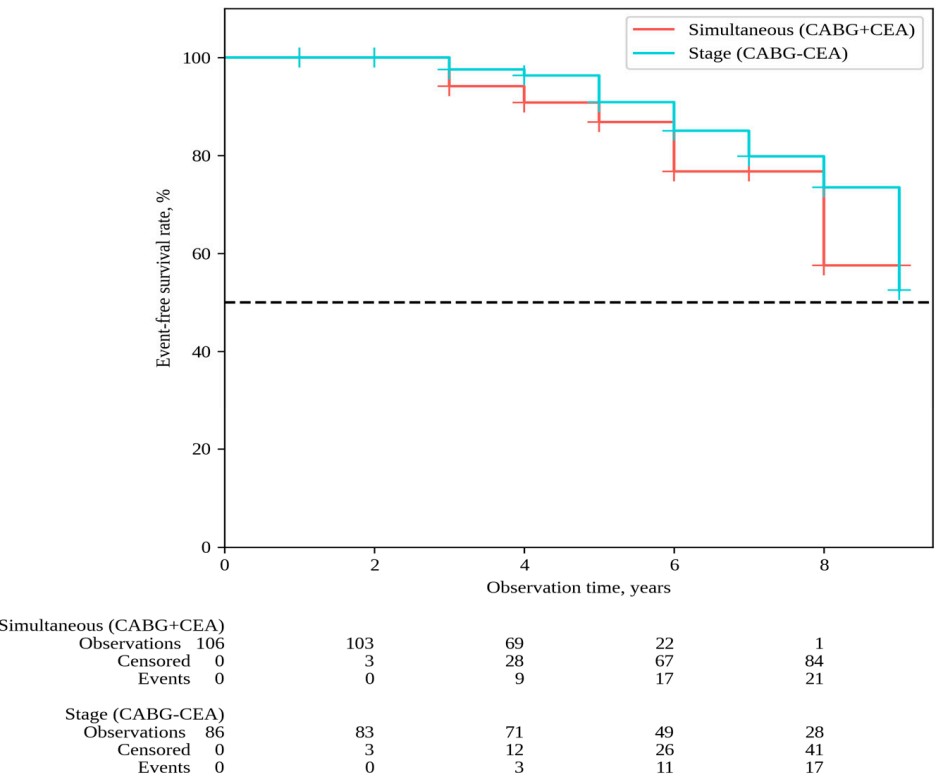

Figure 3. MI-free survival.

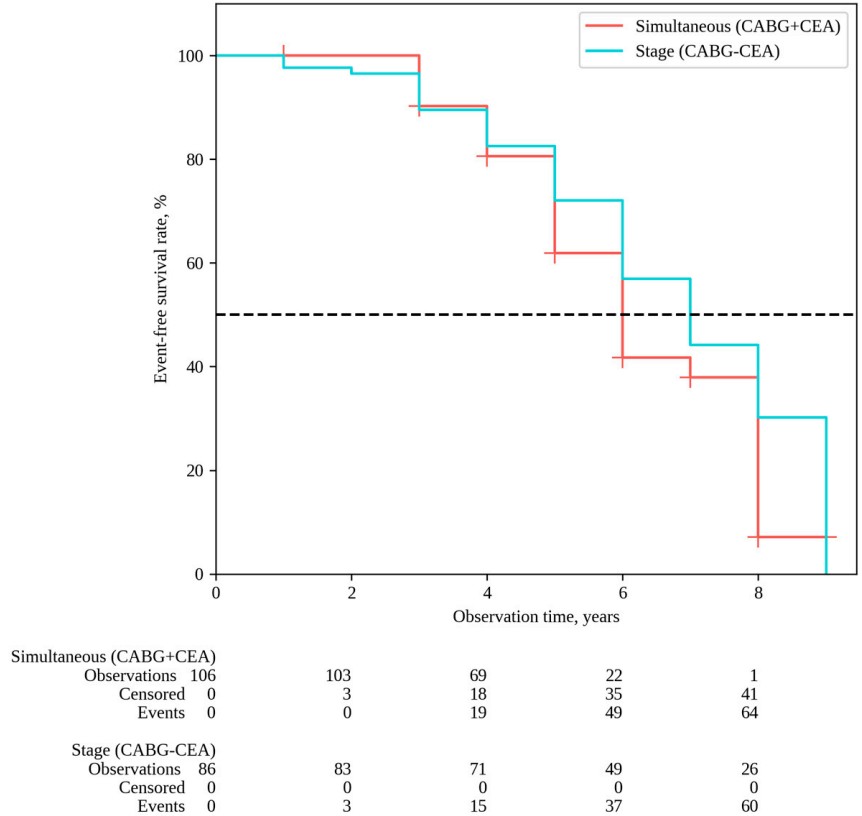

**Figure 4.** MACE-free survival.

**Table 4.** Results of Cox regression analysis to assess risk factors for all-cause mortality.

| Risk Factors | Univariate Analysis | | Multivariate Analysis | |
|---|---|---|---|---|
| | HR (95% CI) | *p*-Value | HR (95% CI) | *p*-Value |
| Age per 1 y | 1.01 (0.97–1.04) | 0.850 | - | - |
| BMI per increase 1 kg/m$^2$ | 0.92 (0.86–0.99) | 0.035 * | 0.93 (0.85–0.99) | 0.036 * |
| Smoking | 1.87 (1.10–3.18) | 0.021 * | - | - |
| Female | 1.47 (0.82–2.64) | 0.198 | - | - |
| Previous stroke | 1.34 (0.72–2.49) | 0.348 | - | - |
| Previous MI | 1.01 (0.56–1.68) | 0.988 | - | - |
| Bilateral CA stenosis | 1.01 (0.43–2.35) | 0.979 | - | - |
| LV EF | 1.03 (0.91–1.17) | 0.621 | - | - |
| Three-vessel disease | 0.53 (0.31–0.98) | 0.063 | - | - |
| LCA trunk disease > 50% | 1.34 (0.79–2.28) | 0.282 | - | - |
| Lower extremity artery stenosis | 0.85 (0.31–2.35) | 0.758 | - | - |
| Diabetes | 0.84 (0.48–1.50) | 0.566 | - | - |
| CKD | 1.34 (0.73–2.44) | 0.341 | - | - |
| COPD | 1.56 (0.62–3.91) | 0.341 | - | - |
| CABG CPB | 1.47 (0.83–2.62) | 0.186 | - | - |
| Stage | 0.62 (0.37–1.04) | 0.069 | - | - |

*—differences in indicators are statistically significant ($p < 0.05$). MACEs—major adverse cardiac events, BMI—body mass index, MI—myocardial infarction, CA—carotid artery, LV EF—left ventricular ejection fraction, LCA—left coronary artery, CKD—chronic kidney disease, COPD—chronic obstructive pulmonary disease, CABG—coronary artery bypass grafting, CPB—cardiopulmonary bypass.

BMI and smoking were associated with the risk of distant MACEs onset in the univariate analysis; nevertheless, in the multivariate analysis, both factors became non-significant (Table 5).

**Table 5.** Results of Cox regression analysis to assess risk factors for MACEs.

| Risk Factors | Univariate Analysis | | Multivariate Analysis | |
|---|---|---|---|---|
| | HR (95% CI) | *p*-Value | HR (95% CI) | *p*-Value |
| Age per 1 y | 1.01 (0.98–1.03) | 0.814 | - | - |
| Female | 1.14 (0.76–1.70) | 0.523 | - | - |
| BMI per 1 kg/m$^2$ | 0.95 (0.91–0.99) | 0.042 * | 0.96 (0.92–1.01) | 0.080 |
| Smoking | 1.44 (1.01–2.04) | 0.043 * | 1.36 (0.95–1.94) | 0.09 |
| Previous Stroke | 1.12 (0.74–1.70) | 0.583 | - | - |
| Previous MI | 0.92 (0.66–1.29) | 0.629 | | |
| Bilateral CA stenosis | 0.96 (0.55–1.67) | 0.896 | - | - |
| LV EF | 0.94 (0.87–1.02) | 0.135 | - | - |
| Three-vessel disease | 0.76 (0.52–1.12) | 0.164 | - | - |
| LCA trunk disease > 50% | 1.19 (0.84–1.68) | 0.331 | - | - |
| Lower extremity artery stenosis | 1.08 (0.60–1.95) | 0.803 | - | - |
| Diabetes | 0.82 (0.57–1.19) | 0.820 | - | - |
| CKD | 1.35 (0.92–1.97) | 0.124 | - | - |
| COPD | 1.40 (0.76–2.59) | 0.283 | - | - |
| CABG CPB | 1.64 (1.12–2.28) | 0.082 | - | - |
| Stage | 0.715 (0.50–1.01) | 0.060 | - | - |

*—differences in indicators are statistically significant ($p < 0.05$). MACEs—major adverse cardiac events, BMI—body mass index, MI—myocardial infarction, CA—carotid artery, LV EF—left ventricular ejection fraction, LCA—left coronary artery, CKD—chronic kidney disease, COPD—chronic obstructive pulmonary disease, CABG—coronary artery bypass grafting, CPB—cardiopulmonary bypass.

In the univariate Cox regression analysis, parameters such as COPD, LV EF, and the presence of critical lesions in the lower extremity arteries were associated with the risk of MI. However, in the multivariate Cox regression analysis, COPD (HR 2.89, 95% CI 1.17–7.13, $p$ = 0.021) and lower extremity arterial lesions (HR 2.45, 95% CI 1.20–7.23, $p$ = 0.018) emerged as the risk factors of MI (Table 6).

**Table 6.** Results of Cox regression analysis for the assessment of MI risk factors.

| Risk Factors | Univariate Analysis | | Multivariate Analysis | |
|---|---|---|---|---|
| | HR (95% CI) | *p*-Value | HR (95% CI) | *p*-Value |
| Age per 1 y | 0.99 (0.95–1.04) | 0.899 | - | - |
| Female | 0.81 (0.34–1.94) | 0.638 | - | - |
| BMI per 1 kg/m$^2$ | 1.03 (0.95–1.11) | 0.510 | - | - |
| Smoking | 1.71 (0.88–3.35) | 0.114 | - | - |
| Previos Stroke | 1.08 (0.47–2.46) | 0.851 | - | - |
| Previous MI | 0.99 (0.32–1.26) | 0.194 | - | - |
| Bilateral CA stenosis | 1.06 (0.38–0.99) | 0.913 | - | - |
| LV EF | 0.84 (0.73–0.98) | 0.022 * | 0.87 (0.74–1.01) | 0.061 |

**Table 6.** *Cont.*

| Risk Factors | Univariate Analysis | | Multivariate Analysis | |
|---|---|---|---|---|
| | HR (95% CI) | *p*-Value | HR (95% CI) | *p*-Value |
| Three-vessel disease | 0.80 (0.38–1.70) | 0.568 | - | - |
| LCA trunk disease > 50% | 0.74 (0.35–1.58) | 0.442 | - | - |
| Lower extremity artery stenosis | 2.24 (0.94–5.37) | 0.069 | 2.45 (1.20–7.23) | 0.018 * |
| Diabetes | 0.94 (0.46–1.89) | 0.856 | - | - |
| CKD | 0.94 (0.42–2.15) | 0.892 | - | - |
| COPD | 3.19 (1.33–7.66) | 0.009 * | 2.89 (1.17–7.13) | 0.021 * |
| CABG CPB | 1.93 (0.48–1.81) | 0.839 | - | - |
| Stage | 0.64 (0.34–1.21) | 0.169 | - | - |

\*—differences in indicators are statistically significant ($p < 0.05$). MI—myocardial infarction, BMI—body mass index, CA—carotid artery, LV EF—left ventricular ejection fraction, LCA—left coronary artery, CKD—chronic kidney disease, COPD—chronic obstructive pulmonary disease, CABG—coronary artery bypass grafting, CPB—cardiopulmonary bypass.

The Cox regression analysis did not identify risk factors associated with the onset of stroke during the long-term follow-up period (Table 7).

**Table 7.** Results of Cox regression analysis for the assessment of stroke risk factors.

| Risk Factors | Univariate Analysis | |
|---|---|---|
| | HR (95% CI) | *p*-Value |
| Age per 1 y | 1.01 (0.96–1.04) | 0.879 |
| Female | 0.88 (0.41–1.88) | 0.745 |
| BMI per 1 kg/m$^2$ | 0.96 (0.89–1.04) | 0.365 |
| Smoking | 0.82 (0.42–1.63) | 0.577 |
| Previous stroke | 0.92 (0.43–1.97) | 0.829 |
| Previous MI | 1.08 (0.61–1.92) | 0.780 |
| Bilateral CA stenosis | 0.83 (0.30–2.30) | 0.715 |
| LV EF | 0.90 (0.79–1.03) | 0.121 |
| Three-vessel disease | 1.30 (0.61–2.78) | 0.500 |
| LCA trunk disease > 50% | 1.42 (0.79–2.56) | 0.236 |
| Lower extremity artery stenosis | 0.80 (0.95–2.58) | 0.710 |
| Diabetes | 0.87 (0.46–1.64) | 0.665 |
| CKD | 1.56 (0.83–2.96) | 0.169 |
| COPD | 0.05 (0.01–10.74) | 0.268 |
| CABG CPB | 2.02 (0.86–4.06) | 0.115 |
| Stage | 0.772 (0.443–1.347) | 0.364 |

BMI—body mass index, MI—myocardial infarction, CA—carotid artery, LV EF—left ventricular ejection fraction, LCA—left coronary artery, CKD—chronic kidney disease, COPD—chronic obstructive pulmonary disease, CABG—coronary artery bypass grafting, CPB—cardiopulmonary bypass.

## 5. Discussion

Systemic atherosclerosis is a prevalent pathology contributing to mortality, with half of cardiovascular deaths linked to CHD and a quarter to ischemic stroke [9]. It is recognized that 20% of patients with confirmed CHD also present with severe lesions in the carotid

arteries; concurrently, up to half of individuals with critical carotid artery stenoses exhibit clinical manifestations of CHD [10].

The aim of a combined surgical intervention on the coronary and carotid arteries is to mitigate the risk of stroke and myocardial infarction resulting from stenosis in the corresponding vascular system. Concurrently, it is established that CABG is associated with a reduced risk of myocardial infarction and cardiovascular death, presenting an advantage over coronary stenting [11]. Furthermore, CABG is known to be accompanied by an increased risk of stroke in approximately 4% of cases [12,13] and may reach up to 9% in patients over 80 years old [2], posing a significant challenge to the postoperative course and prognosis. Notably, up to 25% of strokes in patients with CHD following CABG are attributed to carotid artery lesions [14]. Hence, revascularization of the carotid system in high-risk patients undergoing CABG could be both clinically and economically justified.

The necessity of open surgical revascularization of the coronary and carotid vascular systems is a topic of primary consideration in patients with concomitant severe lesions of the coronary and carotid arteries, which are not amenable to stenting. In such cases, the feasibility of performing CABG and CEA is explored, and this can be accomplished through either simultaneous or staged interventions.

The indications for surgical intervention in the context of concurrent coronary and carotid artery disease have undergone multiple revisions; nevertheless, consensus on the optimal sequence of revascularization remains elusive [7]. The recommendations provided lack a high level of evidence, given the limited number of randomized trials, with only one recommendation attaining the highest level of evidence, emphasizing the necessity for the consilium of a multidisciplinary team to determine the appropriate revascularization strategy.

The current guidelines [7] do not recommend surgical intervention for patients with asymptomatic critical stenosis of 70–99% referred for CABG (class III, level B). However, carotid revascularization in asymptomatic patients undergoing CABG may be considered in the presence of risk factors for ipsilateral stroke to reduce the risk of its occurrence in the postoperative period (class IIb, level C) [15]. In turn, carotid revascularization is recommended (class II A, level B) in patients referred for CABG in the presence of symptomatic carotid lesions in the range of 50–99% [7].

It is acknowledged that symptomatic carotid lesions serve as a significant prognostic factor [7]. Nevertheless, certain authors have reported an increased risk of MI in symptomatic patients undergoing staged interventions [3,16]. In our study, the majority of patients referred for surgical intervention were asymptomatic. However, the existence of symptomatic carotid artery disease (history of stroke) did not emerge as a risk factor associated with 30-day MACEs and distant endpoints (death, MI, stroke, MACEs) according to logistic regression. This might be attributed to the limitations posed by the sample size.

Additionally, it is crucial to emphasize the lack of consensus in the guidelines regarding the indications for a simultaneous or staged intervention, the sequence of stages, and the optimal timing between stages. A randomized trial by Illuminati et al. [17] demonstrated that simultaneous CEA with CABG can more effectively prevent stroke in patients with asymptomatic unilateral carotid stenosis (>70%) than carotid revascularization performed after CABG. The advantages of simultaneous interventions have also been reported in other studies [4,18].

In a meta-analysis by Tzoumas [5] comprising eleven studies involving 44,895 patients (21,710 undergoing simultaneous surgery and 23,185 in the staged surgery group), the simultaneous approach was found to be associated with a higher risk of 30-day mortality (OR 1.33, 95% CI 1.01–1.75, I2 = 47.8%) and stroke (OR 1.51, 95% CI 1.34–1.71, I2 = 0%), but a lower risk of MI (OR 0.15, 95% CI 0.04–0.61, I2 = 0%) compared with the staged approach. Similar conclusions have been drawn by other researchers [6].

On the other hand, a meta-analysis conducted by Sharma et al. [19] assessing surgical outcomes in 17,469 patients following simultaneous intervention and 7552 after staged intervention revealed no significant differences in early mortality, postoperative stroke incidence, and overall scoring between the two surgical strategies.

The results of our study indicated that both interventions demonstrated comparable profiles concerning the postoperative risks of death, MI, stroke, and MACEs. However, individuals undergoing the simultaneous intervention were more prone to prolonged ventilation ($p = 0.004$) and exhibited a higher risk of postoperative complications per patient (OR 2.214, 95% CI 1.048–4.674, $p = 0.035$) when contrasted with the staged approach. Notably, this increased risk did not translate into immediate elevations in the risks of death or MACEs. Importantly, in patients referred for staged surgery, the extended interval between stages (mean 1.8 months) did not result in an elevated risk of complications in the treated vascular system during the second stage, consistent with findings from other studies [5,20].

In practice, the decision regarding the staging and timing of surgical interventions is made not only based on clinical recommendations but also on local protocols, taking into account the capabilities and experience of the clinic. Typically, a simultaneous intervention is preferable when there is a mutually high risk of perioperative MI and stroke, which may be attributed to both the severity of coronary and carotid artery lesions and the low vascular and perfusion reserve of the myocardium and brain. However, the prolonged duration and traumatic nature of simultaneous operations may increase the risk of perioperative complications. In contrast, a staged intervention is characterized by a shorter duration but is more expensive and may be associated with complications from a lower-priority vascular system, the surgical treatment of which is scheduled for the second stage [21].

In our study, age was identified as an independent risk factor for increased hospital MACEs in the overall patient group. The prognostic significance of age in patients with systemic atherosclerosis has been previously suggested [22]. Other risk factors identified in our study, which align with findings from other authors, include smoking [23], COPD [24], decrease of LV EF [25], and atherosclerotic lesions of the lower extremities [26]. Equally intriguing is the "obesity paradox" observed among cardiac surgery patients [27], a trend that was also noted in our study. An increase in BMI was associated with a lower risk of long-term all-cause mortality according to multivariate logistic analysis, as well as 30-day and long-term MACEs.

There are few studies on the long-term outcomes of simultaneous and staged interventions. A study by Haywood et al. [3] investigating long-term outcomes indicated that the adjusted risks of stroke and death were comparable between the simultaneous and staged intervention groups, but the risk of long-term MI was higher in the staged intervention group (OR 1.49; 95% CI, 1.07–2.08; $p = 0.02$). However, another study [28] assessing the immediate and five-year outcomes of two surgical strategies found no differences based on the type of intervention.

In our study, the analysis of long-term outcomes (up to a maximum of 9 years, median—6 years) revealed no significant differences in event-free survival concerning death, stroke, MI, and MACEs between groups, regardless of the approach to CABG and CEA. The observed trend, suggesting the superiority of staged interventions over simultaneous interventions, did not reach statistical significance when evaluating overall survival ($p = 0.068$) and MACE-free survival ($p = 0.060$), aligning with findings reported by other researchers [28].

In general, the literature analysis revealed a lack of consensus regarding the selection of optimal intervention tactics. This discrepancy is attributed to various factors, including (1) the significant heterogeneity among groups (stemming from evolving indications for intervention, varying initial severity of patients' clinical conditions, etc.); (2) variations in the sequence and timing of stages; and (3) diverse surgical approaches and risk profiles among cardiac surgical hospitals, making result comparisons challenging. The implementation of contemporary randomized clinical trials could assist in formulating algorithms for determining optimal surgical tactics.

In our view, when determining the selection between simultaneous or staged intervention tactics for a patient with coronary and carotid artery disease, adopting a patient-centered approach to mitigate immediate postoperative risks is imperative. The choice of

tactics should be based on determining the priority vascular system, assessing anatomical features of both vascular systems, the potential for comprehensive myocardial revascularization, and adequate surgical correction of carotid artery stenosis. When opting for a staged tactic, it involves determining the optimal time between stages and evaluating individual risk factors.

The long-term prognosis, in our view, relies heavily on the patient's clinical condition and accompanying pathology. It is significantly influenced by the adequacy of the administered drug therapy and the attainment of treatment targets. The correlation between the outcomes of myocardial revascularization and medication treatment has been previously noted [29].

Our study has some limitations that do not allow its generalization to all patients with critical unilateral carotid and multiple coronary lesions: (1) It was a single-center retrospective study. A multicenter study will effectively increase the number of cases and increase the statistical power. (2) The limitation of the sample size, leading to an uninformative subgroup analysis. (3) The extended period (from 2013 to 2018) of data collection, accompanied by significant changes in standards of practice and revisions to the indications for surgical intervention. (4) The evolution and revisions in approaches to drug treatment of patients with systemic atherosclerosis. (5) The decrease in clinical significance research on the background of the evolution of endovascular technologies used for revascularization of the coronary and carotid arteries. The active introduction of hybrid revascularization methods into practice dictates the need to compare the long-term results of open interventions with hybrid ones. (6) The lack of ability to differentially diagnose intraoperative and postoperative stroke in patients who experienced a stroke in the perioperative period. (7) The inability to assess the optimality of drug treatment at the time of surgery and during the study period.

## 6. Conclusions

Our study revealed no significant difference in the impact of simultaneous CABG + CEA or staged CABG/CEA interventions on the incidence of death, stroke, MI, and MACEs during a 30-day and long-term follow-up. However, the overall risk of postoperative complications was increased in the group after single-stage interventions compared with the staged intervention group. The duration of lung ventilation was also higher with simultaneous interventions, which is a potentially serious predictor of an unfavorable postoperative course. In our study, these complications did not increase the number of fatal complications after a single-stage intervention, which may be due to a sample limitation. Therefore, the implementation of both tactics is considered appropriate after a thorough operative risk assessment. Randomized clinical trials will be crucial to clarify the optimal intervention tactics based on the specific clinical condition of the patient.

**Author Contributions:** Conceptualization, E.G. and I.S. (Igor Sigaev); methodology, E.G., I.S. (Igor Sigaev) and M.K.; software, B.B. and I.S. (Inessa Slivneva); formal analysis, B.B., N.S. and I.S. (Inessa Slivneva); data curation, M.K., I.V., N.S. and I.S. (Inessa Slivneva); writing—original draft preparation, M.K. and O.K.; writing—review and editing, I.S. (Inessa Slivneva), O.K., V.A. and M.K.; supervision, T.Z. and S.A. All authors have read and agreed to the published version of the manuscript.

**Funding:** This research received no external funding.

**Institutional Review Board Statement:** The study was conducted in accordance with the Declaration of Helsinki, and approved by the local Ethics Committee of Bakulev Scientific Center for Cardiovascular Surgery (approval No. 4/07 November 2023).

**Informed Consent Statement:** Informed consent was obtained from all subjects involved in the study.

**Data Availability Statement:** Data is unavailable due to privacy restrictions.

**Conflicts of Interest:** The authors declare no conflicts of interest.

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
