# Peer review of "Early and Long-Term Results of Simultaneous and Staged Revascularization of Coronary and Carotid Arteries"

_pathophysiology, doi:10.3390/pathophysiology31020017_

Round 1

Reviewer 1 Report

Comments and Suggestions for Authors

Zelikovna et al present the results from a single-center retrospective study of 192 patients with coronary artery disease and carotid artery stenosis ≥70%. They compared two groups of patients - those that underwent simultaneous intervention (CABG+CEA), and others with staged CABG/CEA. At 30-day all-cause mortality, rates of postoperative nonfatal MI, nonfatal stroke, and MACE did not exhibit differences between the groups after single-stage and staged interventions. However, the overall risk of postoperative complications and the duration of ventilatory support were elevated in the group after simultaneous interventions compared with the staged intervention group.  

The study is well-written and presents interesting data. I have a few small comments.

1. In the abstract two consecutive sentences start with "However". I think it will be better to rephrase it.

2. Is the SYNTAX score of the patients available? I think it is important to add to the results of the paper.

Author Response

Response to Reviewer #1.

 The team of authors expresses its deep gratitude for the positive feedback, for very careful and detailed review of our manuscript. Corrections have been made. Regarding the Syntax Score, these data were not obtained in all patients. One of the conditions for inclusion of patients in the study was the completeness of the data analysed. In addition, the initial design of our study was to investigate clinical outcomes and major events of MACE. We analysed the incidence of repeat revascularisation in the distant postoperative period; however, this parameter was not an endpoint in our design. Therefore, we would prefer not to make these changes in the presented study design.

Reviewer 2 Report

Comments and Suggestions for Authors

Suggestions: 

1. This is a single-center, nonrandomized, retrospective study of a relatively low case number (total 192 cases). The strength of the evidence is limited. 

2. The number of patients observed for long term, particularly for the simultaneous intervention group, is extremely low (N = 22 at 6-year, and only 1 at 8-year). The low sample number makes the interpretation  less reliable.

3. The study is to compare the safety and outcomes between simultaneous and staged coronary artery bypass grafting (CABG) and carotid endarterectomy (CAE) for patients with significant carotid a. stenosis on top of significant coronary heart disease requiring intervention on both sites. However, this study did not include percutaneous coronary intervention (PCI), carotid artery stenting (CAS), or both. The clinical outcome of CAS has improved after the launch of the embolic protection device which effectively reduces the peri-CAS embolization risk. 

The combination of CAS with either CABG or PCI, or both, also has no additional concern about the bleeding risk of antiplatelet agents on CEA.

4. The results do not fully support the conclusion. The simultaneous CABG and CAE on overall complications should be addressed.  

References:

Feldman DN, Swaminathan RV, Geleris JD, et al. Comparison of trends and in-hospital outcomes of concurrent carotid artery revascularization and coronary artery bypass graft surgery: The United States Experience 2004 to 2012. JACC Cardiovasc Interv. 2017;10:286-298.)

Tomai F, Pesarini G, Castriota F, et al. Early and long-term outcomes after combined percutaneous revascularization in patients with carotid and coronary artery stenoses. JACC Cardiovasc Interv. 2011;4:560-568. 

Author Response

Response to Reviewer #2.

We express our gratitude for your thorough and meticulous review of our manuscript, which has contributed significantly to its improvement

Suggestions:

  1. This is a single-center, nonrandomized, retrospective study of a relatively low case number (total 192 cases). The strength of the evidence is limited.

The final analysis comprised 192 patients with a complete dataset who underwent surgery within a defined time period. This study is retrospective, providing a cross-sectional profile of data over a specific time span, and the nature of the study design does not allow us to influence the sample size.

  1. The number of patients observed for long term, particularly for the simultaneous intervention group, is extremely low (N = 22 at 6-year, and only 1 at 8-year). The low sample number makes the interpretation less reliable.

We acknowledge the elevated dropout rate among patients, particularly notable in the single-stage surgery group by the sixth year of follow-up. In response to this, we conducted a time-to-event analysis, which considers both the incidence of an event and the timing of its occurrence, while accounting for censored observations. Specifically, we utilized Kaplan-Meier curves to estimate the survival function, conducted a log-rank test to compare groups, and employed Cox regression analysis to evaluate the impact of factors on the outcome.

  1. The study is to compare the safety and outcomes between simultaneous and staged coronary artery bypass grafting (CABG) and carotid endarterectomy (CAE) for patients with significant carotid a. stenosis on top of significant coronary heart disease requiring intervention on both sites. However, this study did not include percutaneous coronary intervention (PCI), carotid artery stenting (CAS), or both. The clinical outcome of CAS has improved after the launch of the embolic protection device which effectively reduces the peri-CAS embolization risk.

The combination of CAS with either CABG or PCI, or both, also has no additional concern about the bleeding risk of antiplatelet agents on CEA.

Certainly, the use of endovascular techniques (including hybrid interventions) in individuals with systemic atherosclerosis has led to a decrease in surgical risk, with comparable long-term outcomes (provided the intervention indications are accurately defined). However, our study aimed to examine the long-term outcomes of the surgical approach to carotid and coronary artery revascularization. Patients who underwent endovascular revascularization were excluded from the study due to the inclusion criterion being open surgery on the coronary and carotid arteries.

  1. The results do not fully support the conclusion. The simultaneous CABG and CAE on overall complications should be addressed.

We acknowledge your feedback and have revised the conclusions accordingly.

Upon analyzing the immediate outcomes of both staged and single-stage coronary and carotid artery surgeries, we did not observe significant differences in the incidence of major postoperative complications. However, we did identify a prolonged duration of ventilatory support in patients undergoing one-stage operations. While this aspect is noteworthy and may potentially contribute to an elevated risk of hospital-acquired complications, our study did not reveal such an association.

Furthermore, the number of patients experiencing any postoperative complications was higher following single-stage interventions. In the long-term follow-up period, no statistically significant differences were observed between the groups concerning endpoints.

Based on these findings, we conclude that the effectiveness of single-stage and staged tactics is comparable in terms of immediate and long-term survival, MI-free survival, stroke, and MACE.

Nevertheless, clinical decision-making should consider the heightened risk of hospital complications associated with single-stage tactics in coronary and carotid artery surgery.

In light of the results obtained, we cannot assert the same level of safety for both approaches. However, we maintain that both surgical tactics are applicable in practice after a thorough assessment of the risk of postoperative complications. It is crucial to emphasize the necessity for randomized clinical trials to determine the optimal intervention tactics.

Round 2

Reviewer 2 Report

Comments and Suggestions for Authors

In the reply to the reviewer, the author insists on the study design and the limited case number. This may be why there is no statistically significant difference between two-stage and simultaneous CABG and CAE. A multi-center design will effectively increase the case number and enhance the statistical power. The author should at least admit the drawback in the study design. 

The clinical significance of this study is attenuated as the success rates of transcatheter interventions to coronary and carotid arteries increased in recent years. The lack of hybrid intervention group should be mentioned as a weak point in the Discussion section.  

On the other hand, prolonged ventilator use will increase the length of ICU stay, which is a major adverse effect by definition. The author should address this AE in the conclusion section. 

Author Response

In the reply to the reviewer, the author insists on the study design and the limited case number. This may be why there is no statistically significant difference between two-stage and simultaneous CABG and CAE. A multi-center design will effectively increase the case number and enhance the statistical power. The author should at least admit the drawback in the study design. 

Added a mention of this deficiency in the study limitations.

(1) The study limitations include its retrospective, single-center design.

A multicenter study would effectively increase case numbers and enhance statistical power.

The clinical significance of this study is attenuated as the success rates of transcatheter interventions to coronary and carotid arteries increased in recent years. The lack of hybrid intervention group should be mentioned as a weak point in the Discussion section.  

Added acknowledgment of a limitation of the study:

(5) diminished clinical significance research in light of evolving endovascular technologies employed for coronary and carotid artery revascularization. The widespread adoption of hybrid revascularization methods necessitates a comparison of long-term results between open interventions and hybrid approaches. Therefore, the absence of a group undergoing hybrid interventions represents a notable limitation of this study.

On the other hand, prolonged ventilator use will increase the length of ICU stay, which is a major adverse effect by definition. The author should address this AE in the conclusion section. 

Added to the Conclusion.

Conclusion

Our study revealed no significant difference in the impact of simultaneous CABG + CEA or staged CABG/CEA interventions on the incidence of death, stroke, MI, and MACE during the 30-day and long-term follow-up. Nevertheless, the overall risk of postoperative complications was elevated in the group undergoing single-stage interventions compared to the staged intervention group. The duration of lung ventilation was also prolonged in simultaneous interventions, indicating a potentially significant predictor of an adverse postoperative course. In our study, these complications did not result in an increased incidence of fatal complications after single-stage intervention, a limitation that may be attributed to the sample size. Therefore, the implementation of both tactics is considered appropriate after a thorough postoperative risk assessment. Randomized clinical trials will be crucial to clarify the optimal intervention tactics based on the specific clinical condition of the patient.